# Individual history of winning and hierarchy landscape influence stress susceptibility in mice

Katherine B LeClair[1,2,3], Kenny L Chan[1,2], Manuella P Kaster[1,2,4], Lyonna F Parise[1,2], Charles Joseph Burnett[1,2], Scott J Russo[1,2,3]*

[1]Nash Family Department of Neuroscience, New York, United States; [2]Friedman Brain Institute, New York, United States; [3]Graduate School of Biological Science, Icahn School of Medicine at Mount Sinai, New York, United States; [4]Department of Biochemistry, Federal University of Santa Catarina, Santa Catarina, Brazil

**ABSTRACT** Social hierarchy formation is strongly evolutionarily conserved. Across species, rank within social hierarchy has large effects on health and behavior. To investigate the relationship between social rank and stress susceptibility, we exposed ranked male and female mice to social and non-social stressors and manipulated social hierarchy position. We found that rank predicts same sex social stress outcomes: dominance in males and females confers resilience while subordination confers susceptibility. Pre-existing rank does not predict non-social stress outcomes in females and weakly does so in males, but rank emerging under stress conditions reveals social interaction deficits in male and female subordinates. Both history of winning and rank of cage mates affect stress susceptibility in males: rising to the top rank through high mobility confers resilience and mice that lose dominance lose stress resilience, although gaining dominance over a subordinate animal does not confer resilience. Overall, we have demonstrated a relationship between social status and stress susceptibility, particularly when taking into account individual history of winning and the overall hierarchy landscape in male and female mice.

*For correspondence: scott.russo@mssm.edu

Competing interest: The authors declare that no competing interests exist.

## Introduction

The formation of social hierarchies is an evolutionarily conserved phenomenon having been observed across species from insects and fish to rodents, non-human primates, and humans (*Bonabeau et al., 1999*). Position within the hierarchy can determine access to important resources like food, water, and potential mates, with dominant animals gaining preferential access. It has been hypothesized that this predetermination of resource allocation helps to reduce conflicts within the group (*Hausfater et al., 1982*; *Savin-Williams, 1980*; *Schülke et al., 2010*; *Wang et al., 2014*). In humans, one measure of social hierarchy position is socioeconomic status (SES) which is a relative combined measure of an individual's income, education and occupation (*Shavers, 2007*). Strikingly, SES has been identified as one of the single strongest predictors of mortality and morbidity, with individuals from a low SES having increased incidence of heart disease, stroke, bronchitis, and multiple mental health disorders including major depressive disorder as well as increased incidence of psychological distress (*Adler and Ostrove, 1999*; *Calixto and Anaya, 2014*; *Shishehbor et al., 2006*; *Weissman et al., 2015*). This relationship has been seen in multiple countries including those with universal healthcare and when accounting for lifestyle factors associated with SES such as smoking, diet, and physical activity levels (*Groffen et al., 2013*; *Marmot et al., 1991*). While SES is not equivalent to animal rank—human social rank is the consequence of an interplay between multiple factors, including gender, race, education, and wealth—a similar pattern of deleterious health outcomes associated with low rank has

been observed in animals (*Backström and Winberg, 2017*; *Bartolomucci et al., 2005*; *Cavigelli and Caruso, 2015*; *Czoty et al., 2009*; *Fuchs et al., 1995*; *Gruenewald et al., 2006*; *Sapolsky, 2005*; *Schuhr, 1987*; *Sherman and Mehta, 2020*; *Yodyingyuad et al., 1985*). These findings have led to the proposal that differential exposure to psychosocial stress due to one's social standing could lay the foundation for future health risks. Despite this, little is understood about the etiology driving this relationship between social status and health risks.

Primate research has identified internal and external factors mediating the relationship between rank and stress response, including the structure of the social hierarchy and how that structure is maintained as well as the availability of mitigating factors like social support in the forms of affiliation and coalition forming, and the coping behaviors of the animals themselves (*Abbott et al., 2003*; *Blanchard et al., 1993*; *Setchell et al., 2010*; *Vandeleest et al., 2016*). Recent studies in rodents probing the biology underlying the relationship between rank and stress response in mice have garnered conflicting results in the predictive ability of rank on stress susceptibility (*Karamihalev et al., 2020*; *Larrieu et al., 2017*; *Šabanović et al., 2020*). It is likely that similar factors identified in primates also mediate the relationship between social rank and stress response in rodents. Here, we look to characterize the relationship between social rank and stress susceptibility in mice. In particular, we sought to delineate important aspects of both hierarchy formation and maintenance as well as the effects of overall hierarchy landscape on susceptibility to both social and non-social stressors.

## Results

### Social hierarchy characterization in male and female mice

To assess social hierarchy rank and dynamics over time, we tested same sex tetrads in a round robin of pair-wise tube tests following 3.5 weeks of group housing to allow for initial hierarchy formation. We determined the rank position of each mouse for each day across 18 days of testing using the total win number in that day's tube tests and represented that position over time (*Figure 1A*). Male and female social hierarchies had no significant differences in the number of non-linear days, days in which tube tests results were not fully transitive, (*Figure 1B*) or total stable, four consecutive identical tube test results, or unstable social hierarchies assessed on the final four tube test trials (*Figure 1C*).

As we included mice from both stable and unstable social hierarchies, we calculated rank via David's Score (DS) across the final four tube test results, conducted over one week. DS ranges from –6 to 6 with a higher DS associated with greater dominance. DS weights wins from individual pairings relative to the hierarchy position of the opponent, with wins over more dominant animals weighted more than over subordinate animals. Groups were created by dividing the range of DS into three parts. Mice that had a DS within the top 25 % (DS >3) were considered dominant (DOM), mice that were in the bottom 25 % were considered subordinate (SUB) (DS $\leq$ –3), and mice with a DS within the central 50 % of DS (–3< DS < 3) were considered intermediate (INT) (*Figure 1D and E*).

We validated ranks calculated from tube testing with the warm corner test, which has been used previously to assess social rank (*Zhou et al., 2017*). The warm corner test models ethological competition for a valuable resource, in this case a warm corner of a cage with an otherwise ice-cold floor. Both male (*Figure 1E*) and female (*Figure 1F*) mice show significant positive correlations between dominance and time spent in the warm spot. Dominant males spent significantly more time in the warm corner than both intermediate and subordinate mice (*Figure 1G*). Dominant female mice spent significantly more time in the warm corner than subordinates (*Figure 1H*).

### Social hierarchy rank as a predictor of social stress outcomes in male and female Mice

To determine the predictive role of rank on stress susceptibility, we subjected ranked male and female mice to 10 days of chronic social defeat stress (CSDS). Susceptibility was assayed via a social interaction (SI) test where SI ratio was derived from time spent investigating a novel aggressor, normalized to time spent investigating the empty cage prior to aggressor introduction. Under unstressed conditions, mice spend more time interacting with a novel aggressor than an empty but novel enclosure (SI ratio >1). For male mice, we used the established CSDS model (*Figure 2A*); for females, we used two models of female social defeat: one employing male transgenic *Esr1*-Cre aggressor mice, expressing AAV-Gq-DREADD in the ventromedial hypothalamus (*Figure 2B*), and another using territorial female

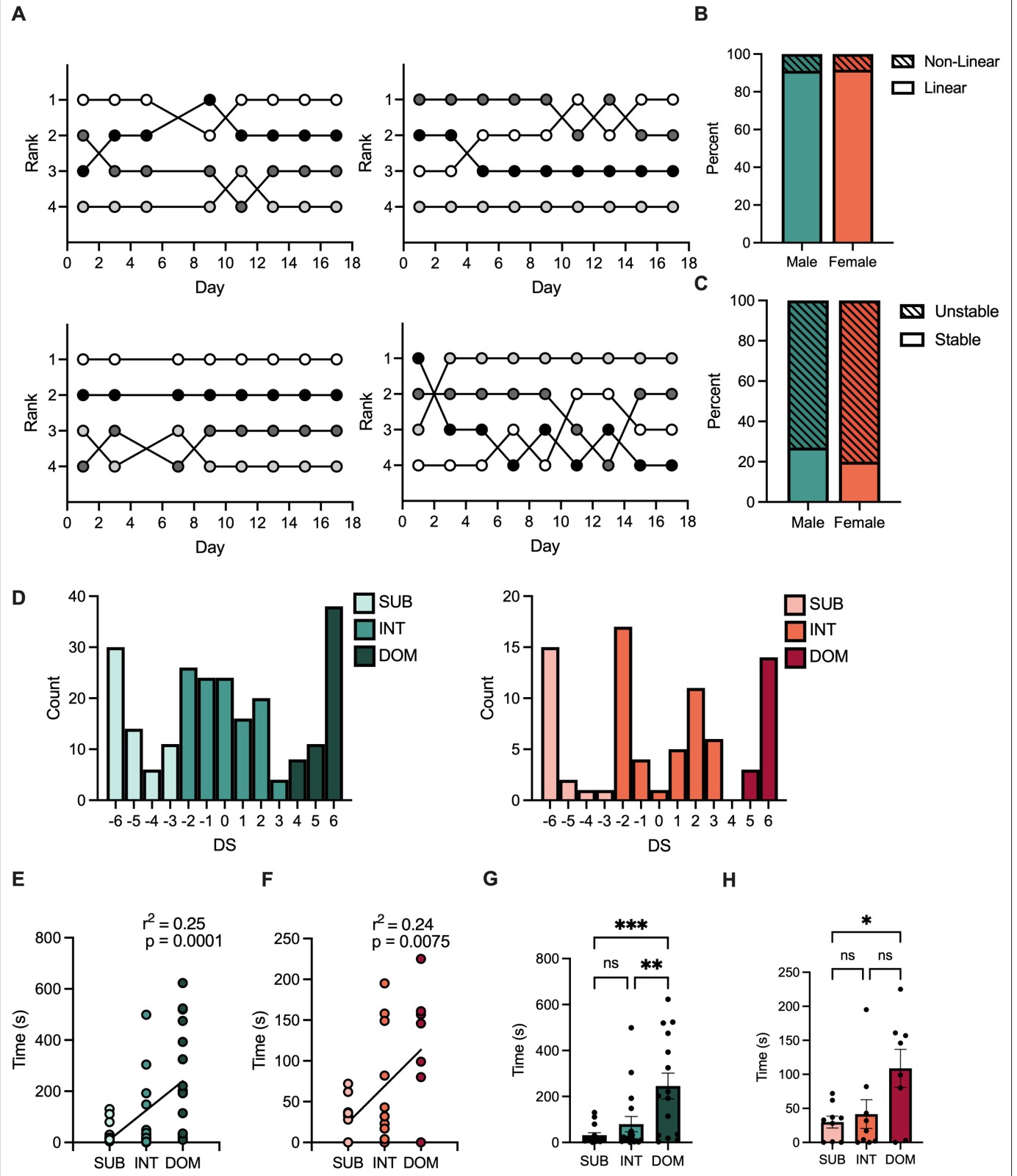

**Figure 1.** Characterization of male and female social hierarchies. (**A**) Representative hierarchies of male (top) and female (bottom) mice over 18 days of hierarchy testing. Each line represents a single animal and its rank position based on summed tube test wins. Non-linear days, days in which tube test results were not fully transitive, are not displayed. (**B**) Proportion of male (teal) and female (orange) cages demonstrating unstable (striped) or stable (solid) hierarchies, defined as four consecutive identical tube test results on the final 8 days of hierarchy testing. $X^2$ (1, N = 2602) = 0.23, p = 0.7429. (**C**)

*Figure 1 continued on next page*

*Figure 1 continued*

Proportion of days that male (teal) and female (orange) cages showed non-linear (striped) or linear (solid) hierarchy results across all days of tube tests. $X^2$ (1, N = 173) = 0.01, p = 0.3170. (**D**) Distribution of David's scores (DS) from male (teal) and female (orange) mice calculated across the final 4 days of tube test results. Subordinate (SUB) = bottom 25 % (DS $\leq$ –3). Dominant (DOM) = top 25 % (DS >3) Intermediate (INT) = central 50 % (-3< DS $\leq$ 3). (**E**) Male rank from tube test plotted against time spent in warm corner (mean ± SEM). Pearson r(26) = 0.25, 95% CI [57.33, 171.9], p < 0.0001 (n = 14–18/group) (**F**) Female rank from tube test plotted against average time spent in warm corner (mean ± SEM). Pearson r(26) = 0.24, 95% CI [12.97, 76.23], p = 0.0075. (n = 7–11/group) (**G**) Time spent in warm spot by rank in males. One-way ANOVA F(2, 44) = 8.60, p = 0.0007. Tukey's Multiple Comparisons. SUB vs INT 95% CI [–176.0, 79.95] p = 0.6369 SUB vs DOM 95% CI [–345.9,–82.15] p = 0.0008, INT vs DOM 95% CI [–294, –38] p = 0.0082 (n = 14–18/group) (**H**) Time spent in warm spot by rank in females. One-way ANOVA F(2, 23) = 4.25, p = 0.0268. Tukey's Multiple Comparisons. SUB vs INT 95% CI [–82.15, 58.59] p = 0.9081 SUB vs DOM 95% CI [-151.5,–6.449] p = 0.0311, INT vs DOM 95% CI [–139.7, 5.329] p = 0.0729 (n = 7–11/group).

The online version of this article includes the following source data for figure 1:

**Source data 1.** David's scores of male and female mice.

Swiss Webster aggressor mice (*Figure 2C*). Twenty-four hours after the final day of stress, mice underwent the social interaction (SI) test.

Both subordinate and intermediate male mice had a significantly lower SI ratio than unstressed controls following CSDS whereas dominant mice did not (*Figure 2D*). We observed a moderate positive correlation between DS and SI Ratio in stressed male mice (*Figure 2E*). In females socially defeated by male mice, however, all ranks show significantly lower SI ratios than unstressed controls (*Figure 2F*), and no significant correlation was seen between DS and SI ratio (*Figure 2G*). To further probe the relationship between rank and stress susceptibility in females, we employed a second model of female social defeat, inter-female CSDS. Following 10 days of inter-female CSDS, a significant effect of stress is seen with only subordinate females showing significantly higher vigilance behaviors than unstressed controls (*Figure 2H*). However, no correlation between vigilance and DS was observed (*Figure 2I*).

To assess the relationship between individual behavioral responses to aggression and stress susceptibility, we calculated an active coping score derived from the number of active and passive behaviors displayed during CSDS normalized to the number of attacks received. Active behaviors were defined as running away from the aggressor after attack initiation, or biting, boxing, or lunging at the aggressor. Passive behaviors included immobility, or limited movement in response to attack initiation. While subordinate and dominant male mice displayed similar levels of active and passive coping behaviors on day 1 of defeat, these behaviors differed significantly by day 10, with dominant animals displaying more active coping behaviors than subordinate animals (*Figure 2J*). Correlating day 10 coping behaviors and SI ratios revealed a moderate positive relationship between coping behavior and SI ratio in males, with more active coping being associated with a more resilient phenotype (*Figure 2K*). Dominant and subordinate females had similar coping scores on days 1 and 10 of female CSDS (*Figure 2L*) with a near-significant trend towards a positive relationship between day 10 coping score and SI ratio (*Figure 2M*). In response to inter-female CSDS, both dominant and subordinate female mice displayed similar levels of coping behaviors (*Figure 2N*) with no correlation between SI ratio and coping score (*Figure 2O*). Moreover, no correlation was seen between coping score and vigilance displayed during the SI test (*Figure 2—figure supplement 1A*).

Additional assessment of behavioral or physiological differences revealed no significant effects across rank. No pre-existing differences were observed in body weight between all ranks of mice before stress in males (*Figure 2—figure supplement 1B*) or females (*Figure 2—figure supplement 1C*). Importantly, in all models of social stress, there were no significant differences in bouts of aggression received by dominant or subordinate mice over the course of the defeat (scored on days 1, 5, and 10 of defeat) in both male (*Figure 2—figure supplement 1D*) and female mice (*Figure 2—figure supplement 1E* and *Figure 2—figure supplement 1F*). Wound score, an indirect measure of aggression taken 48 hr after defeat, revealed no significant differences between dominant and subordinate males (*Figure 2—figure supplement 1G*) or females (*Figure 2—figure supplement 1H* and *Figure 2—figure supplement 1I*). There were also no significant differences in distance moved when the target was absent during SI testing in males (*Figure 2—figure supplement 2A*) or females following CSDS or inter-female CSDS (*Figure 2—figure supplement 2B* and *Figure 2—figure supplement 2C*). Lastly, as previously reported (*Newman et al., 2019*), inter-female CSDS caused no changes in SI ratio between stressed or unstressed controls, and no observable differences across ranks (*Figure 2—figure supplement 2D*).

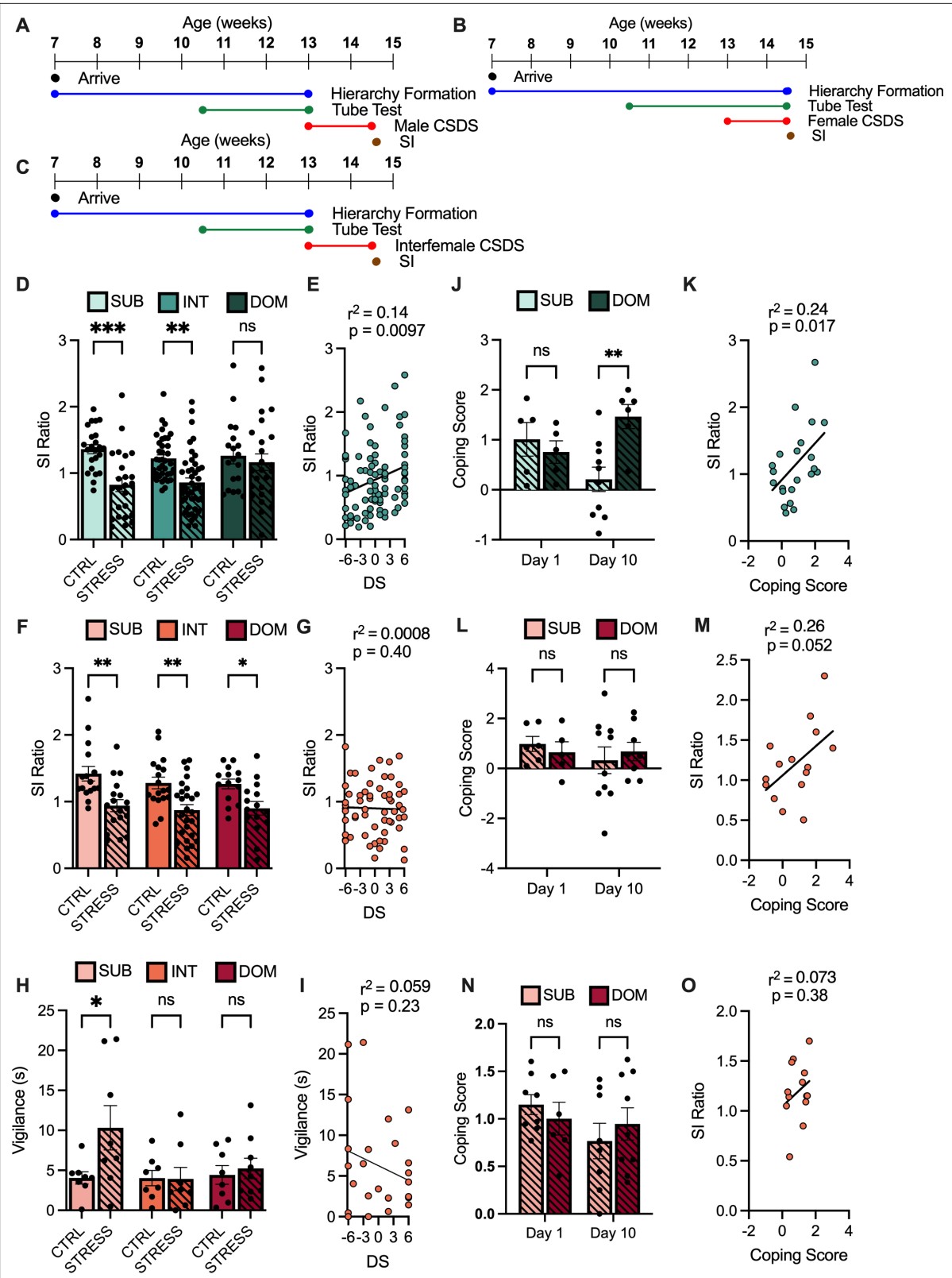

**Figure 2.** Male and female mice show alternate patterns of response to social stress. (**A**) Timeline of chronic social defeat stress (CSDS) in male mice. (**B**) Timeline of CSDS in female mice using male DREADD aggressors. (**C**) Timeline of CSDS in female mice using female aggressors. (**D**) Social interaction (SI) ratio between group-housed unstressed control (CTRL) and stressed (STRESS) male mice (mean ± SEM). Two-way ANOVA Stress $F_{(1, 166)}$ = 20.93, p < 0.0001. Sidak's multiple comparisons of STRESS versus CTRL within groups SUB 95% CI[0.21, 0.85] p = 0.003 INT 95% CI [0.12, 0.61] p = 0.0014 DOM

*Figure 2 continued on next page*

*Figure 2 continued*

95% CI[–0.24,.43] p = 0.863 (n = 19–45/group). (**E**) Correlation between David's score (DS) and SI ratio. Pearson r(89) = 0.14, 95% CI [0.0086, 0.061] p = 0.0097 (n = 91). (**F**) SI ratio between CTRL and STRESS female mice (mean ± SEM). Two-way ANOVA Stress F(1, 101) = 29.62, p < 0.0001. Sidak's multiple comparisons of STRESS versus CTRL within groups SUB 95% CI [0.16, 0.80] p = 0.0015 INT 95% CI [0.12, 0.70], p = 0.0027 DOM 95% CI [0.01, 0.71] p = 0.0392 (n = 16–30/group). (**G**) Correlation between DS and SI Ratio. Pearson r(56) = 0.00002, 95% CI [–0.02, 0.02] p = 0.402 (**H**) Vigilance (seconds) between CTRL and STRESS female mice (mean ± SEM). Two-way ANOVA Stress F(1, 43) = 3.441, p = 0.0704, SUB 95% CI [-11.71,–0.83] p = 0.0193 INT 95% CI [–5.35, 5.55] p > 0.9999 DOM 95% CI [–6.10, 4.48] p = 0.9747 (n = 8–9/group). (**I**) Correlation between DS and vigilance. Pearson r(13) = 0.26, 95% CI [–0.80, 0.20] p = 0.23 (n = 15). (**J**) Coping score from stressed males on day 1 and day 10 of CSDS (mean ± SEM). Two-way ANOVA interaction F(1, 22) = 7.29, 95% CI [-2.67,–0.35] p = 0.0193. Sidak's multiple comparisons of DOM versus SUB within groups Day 1 95% CI [–0.78, 1.29] p = 0.8088 Day 10 95% CI [–2.1, 0.4] p = 0.0036 (n = 5–10/group). (**K**) Correlation between Coping Score and SI Ratio of stressed males. Pearson r(21) = 0.24, 95% CI [0.053, 0.49], p = 0.00174 (n = 23). (**L**) Coping score from stressed females on day 1 and day 10 of CSDS (mean ± SEM). Two-way ANOVA interaction F(1, 25) = .4870, p = 0.3285, (n = 5–10/ group). (**M**) Correlation between Coping Score and SI Ratio of stressed females. Pearson r(13) = 0.26, 95% CI [–0.0016, 0.37] p = 0.052 (n = 15). (**N**) Coping score from stressed animals on day 1 and day 10 of inter-female CSDS (mean ± SEM). Two-way ANOVA interaction F(1, 27) = .99, p = 0.3285 (n = 8–10/group). (**O**) Correlation between Coping Score and SI Ratio. Pearson r(11) = 0.07, 95% CI [–0.21, 0.53], p = 0.3736 (n = 13). For all graphs *p ≤ 0.05, **p ≤ 0.01, ***p ≤ 0.001.

The online version of this article includes the following source data and figure supplement(s) for figure 2:

**Figure supplement 1.** Aggression received during defeat.

**Figure supplement 1—source data 1.** Bouts of attack across defeat models.

**Figure supplement 2.** Additional behaviors.

## Interactions between rank and susceptibility versus resilience to non-social stress

We next investigated whether rank was predictive of stress responses to a chronic non-social stressor. To do this we subjected ranked male and female mice to 21 days of chronic variable stress (CVS) (*Figure 3A*). As male and female mice remain group-housed during CVS we were able to examine the effects of stress on social hierarchy dynamics as well as determine the predictive role of pre-stress rank – DS calculated from the week prior to CVS – versus emergent rank – DS calculated from the final week of CVS.

Qualitatively, unstressed and stressed hierarchy dynamics were similar over time (*Figure 3B*). In both male and female mice, dominant animals' average wins remained stable over time, while the average wins of the lower ranks converged over time. This convergence was due to movement in ranks among the lower ranked animals within a cage, unlike dominant animals who remained on top. Steepness, a measure of the slope of ordered normalized DS of mice within a cage, showed no significant differences before or after CVS in both males (*Figure 3C*) and females (*Figure 3D*). Examination of the percentage of animals changing rank position pre- and post-stress revealed similar patterns of rank movement in control and stressed females (*Figure 3—figure supplement 1A*), while control male cages appeared to have more animals changing rank over the time course of the stress period compared to stressed male cages (*Figure 3—figure supplement 1B*).

Dividing mice into pre-stress ranks revealed no significant differences in SI ratio between unstressed and stressed animals across ranks in both males (*Figure 3E*) and females (*Figure 3G*). We observed a positive correlation between DS and SI ratio in males (*Figure 3F*), but not in females (*Figure 3H*). However, separating animals by rank determined at the end of CVS revealed significantly lower SI ratios in subordinates alone when comparing stressed versus unstressed controls in males (*Figure 3I*) and females (*Figure 3J*). This emergent DS correlated with SI ratio in both males (*Figure 3K*) and females (*Figure 3L*).

## Effects of rank change on stress susceptibility versus resilience

To determine whether rank is causally associated with stress susceptibility or resilience, we recombined previously dominant or subordinate animals forcing one to assume an alternate position (*Figure 4A*). In other words, two previously dominant animals from separate cages were regrouped together and dominance was subsequently tested over a 1-week period. Subordinate animals were tested in a parallel manner. Following this recombination and a week of newly established social hierarchies, we subjected the animals to either CSDS or CVS (*Figure 4B and C*). After CSDS, dominant animals that remained dominant following recombination (DOM - DOM) had a significantly higher SI ratio than dominants that became subordinate (DOM - SUB) (*Figure 4D*). However, there was no significant

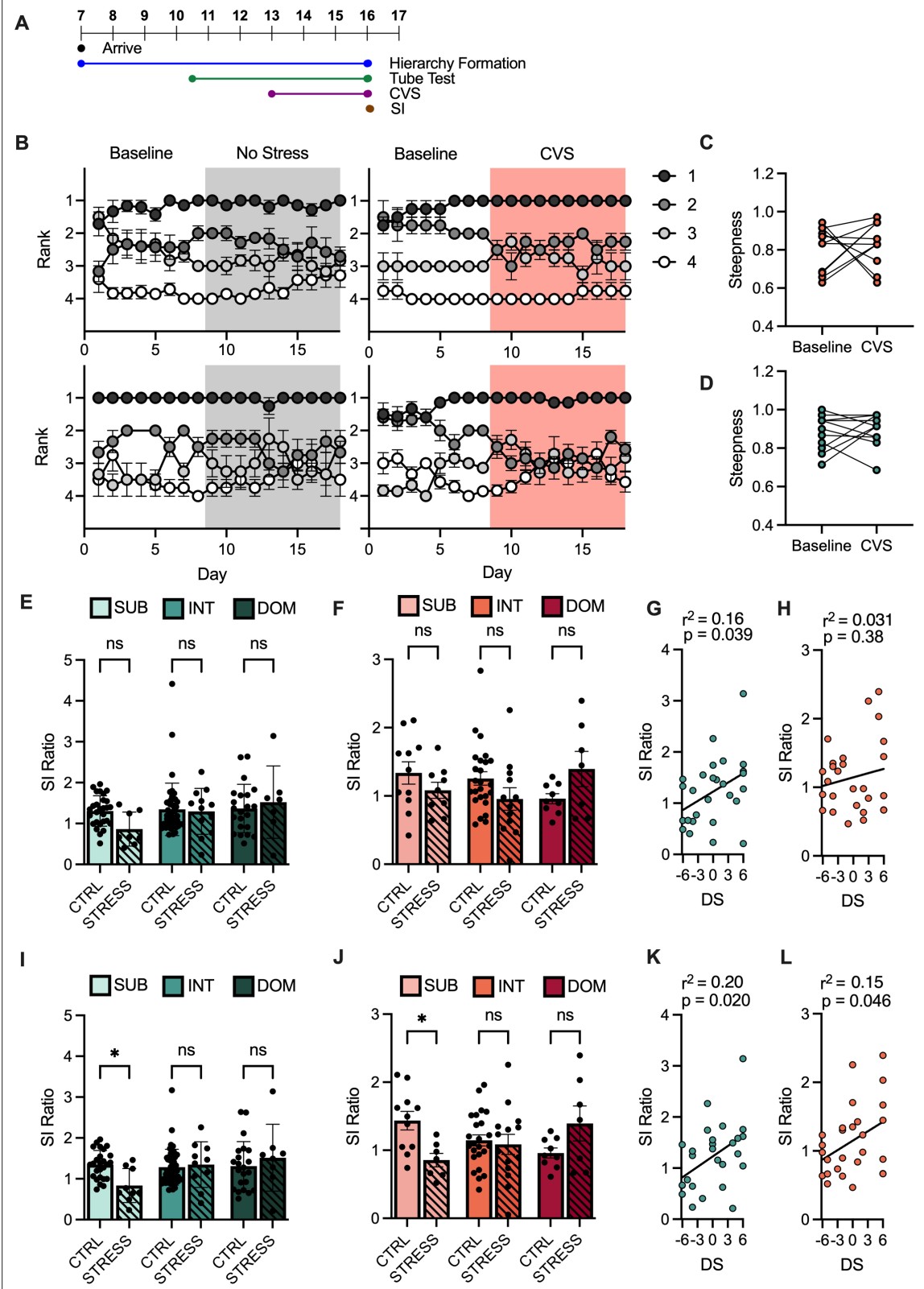

**Figure 3.** Emergent male and female subordinates are susceptible to chronic variable stress (CVS). (**A**) Timeline of chronic variable stress (CVS) experiment. (**B**) Average win number of animals across all days of hierarchy evaluation using David's score (DS) calculated on rank from the 4 days preceding CVS start. (**C**) Male hierarchy steepness calculated from ordered normalized DS across 7 days preceding or following stress initiation. Paired t-test t(10) = 0.3914, p = 0.7037 (n = 11). (**D**) Female hierarchy steepness calculated from ordered normalized DS across 7 days preceding or following

*Figure 3 continued on next page*

*Figure 3 continued*

stress initiation. Paired t-test t(10) = 0.49, 95% CI [–0.10, 0.16], p = 0.6315 (n = 11). (**E**) Social interaction (SI) ratio between unstressed (CTRL) and stressed (STRESS) male mice (mean ± SEM) with rank derived the week preceding CVS. Two-way ANOVA Stress F(1, 116) = 0.75, p = 0.0996 (n = 22–45/group). (**F**) SI ratio between CTRL and STRESS female mice (mean ± SEM) with rank derived the week preceding CVS. Two-way ANOVA Stress F(1, 67) = 0.10, p = 0.7979 (n = 9–25/group) (**G**) Correlation between DS calculated in the week preceding CVS and SI Ratio in males. Pearson r(25) = 0.16, 95% CI [.0032, 0.12], p = 0.0393 (n = 27). (**H**) Correlation between DS calculated in the week preceding CVS and SI Ratio in females. Pearson r(25) = 0.20, 95% CI [–0.027, 0.069], p = 0.3823 (n = 27). (**I**) SI ratio between CTRL and STRESS male mice (mean ± SEM) with rank derived from the final week of stress. Two-way ANOVA interaction of stress and rank F(2, 1150) = 3.89, p = 0.0233. Sidak's multiple comparisons of STRESS versus CTRL within ranks, SUB 95% CI [0.049, 0.98] p = 0.0250 INT 95% CI [–0.48, 0.35] p = 0.9752 DOM 95% CI [–0.70, 0.29] p = 0.6835 (n = 22–46/group). (**J**) SI ratio between CTRL and STRESS female mice (mean ± SEM) with rank derived from the final week of stress. Two-way ANOVA interaction of stress and rank F(2, 66) = 5.27, p = 0.0075. Sidak's multiple comparisons of STRESS versus CTRL within ranks, SUB 95% CI [0.049, 1.11], p = 0.0277 INT 95% CI [–0.31, 0.43] p = 0.9731 DOM 95% CI [–0.99, 0.11] p = 0.1654 (n = 9–24 / group). (**K**) Correlation between DS calculated across final week of CVS and SI Ratio in males. Pearson r(25) = 0.03, 95% CI [0.012, 0.12] p = 0.0195 (n = 27). (**L**) Correlation between DS calculated across final week of CVS and SI Ratio in females. Pearson r(25) = 0.15, 95% CI [0.00096, 0.090] p = 0.0456 (n = 27). For all graphs *p ≤ 0.05, **p ≤ 0.01, ***p ≤ 0.001.

The online version of this article includes the following figure supplement(s) for figure 3:

**Figure supplement 1.** Rank change during Chronic Variable Stress.

difference in SI ratio between subordinates who become dominant (SUB - DOM) versus those that remained subordinate after CSDS (SUB - SUB) (*Figure 4E*). In males, we see a similar pattern of effect of CVS on SI ratio, with DOM - DOM mice having a significantly higher SI ratio than DOM – SUB mice, and no significant differences between SUB - DOM and SUB - SUB animals (*Figure 4F and G*). In females, neither manipulation had an effect on SI ratio following CVS with no significant differences between DOM - DOM and DOM - SUB (*Figure 3H*) or SUB - DOM and SUB - SUB animals (*Figure 3I*).

## Effects of rank mobility on stress susceptibility and resilience

To retrospectively assess the role of individual history of rank in stress susceptibility, we examined rank mobility of male and female mice preceding CSDS. Mice were separated into two groups, those that experienced high mobility, with two or greater changes in rank across all hierarchy testing days preceding stress, and those with low mobility, one or fewer changes in rank across hierarchy testing days preceding stress (*Figure 4J*). Male and female mice displayed similar rates of high and low mobility across testing days (*Figure 4K*). High mobility dominant male mice had a significantly higher SI ratio following stress than low mobility dominant mice, while no significant differences were seen between low and high mobility at any other ranks (*Figure 4L*). No differences between low and high mobility animals across ranks were observed in females following CSDS (*Figure 4M*).

## Discussion

In this study, we sought to probe the role of rank as a predictor for stress response. To do this, we subjected male and female mice from established social ranks to either social (CSDS) or non-social (CVS) stress then evaluated social interaction and coping responses. Next, we manipulated rank by forcing established subordinate or dominant animals to assume a new rank position by recombining them with novel animals with similar rankings. We find that dominant animals, both male and female, are resilient to social stress but subordinates are susceptible to social defeat stress only when it is performed with an aggressor of the same sex. We find that rank is not predictive of stress responses to non-social stressors, but the rank which forms under non-social stress conditions is. These results highlight the ability for rank to serve as a predictor for stress outcomes in a stress-specific manner. This is consistent with the proposal that the stress of social defeat is associated with loss of rank and not merely exposure to the agonistic encounter (*Larrieu and Sandi, 2018*). Our findings also suggest that social rank viewed as an equivalent measure from cage to cage is likely insufficient to properly capture the predictive ability of rank; instead, both history of winning as well as the hierarchy landscape when forming rank interact to impact stress outcomes.

Previous work has revealed multiple sex specific differences in hierarchy formation. Examination of agonistic behavior in a semi-naturalistic environment reveals that male social hierarchies are steeper, more linear, and more despotic than female hierarchies (*Karamihalev et al., 2020*; *Williamson et al., 2019*). These results align with data indicating that male and female mice rely on extrinsic versus

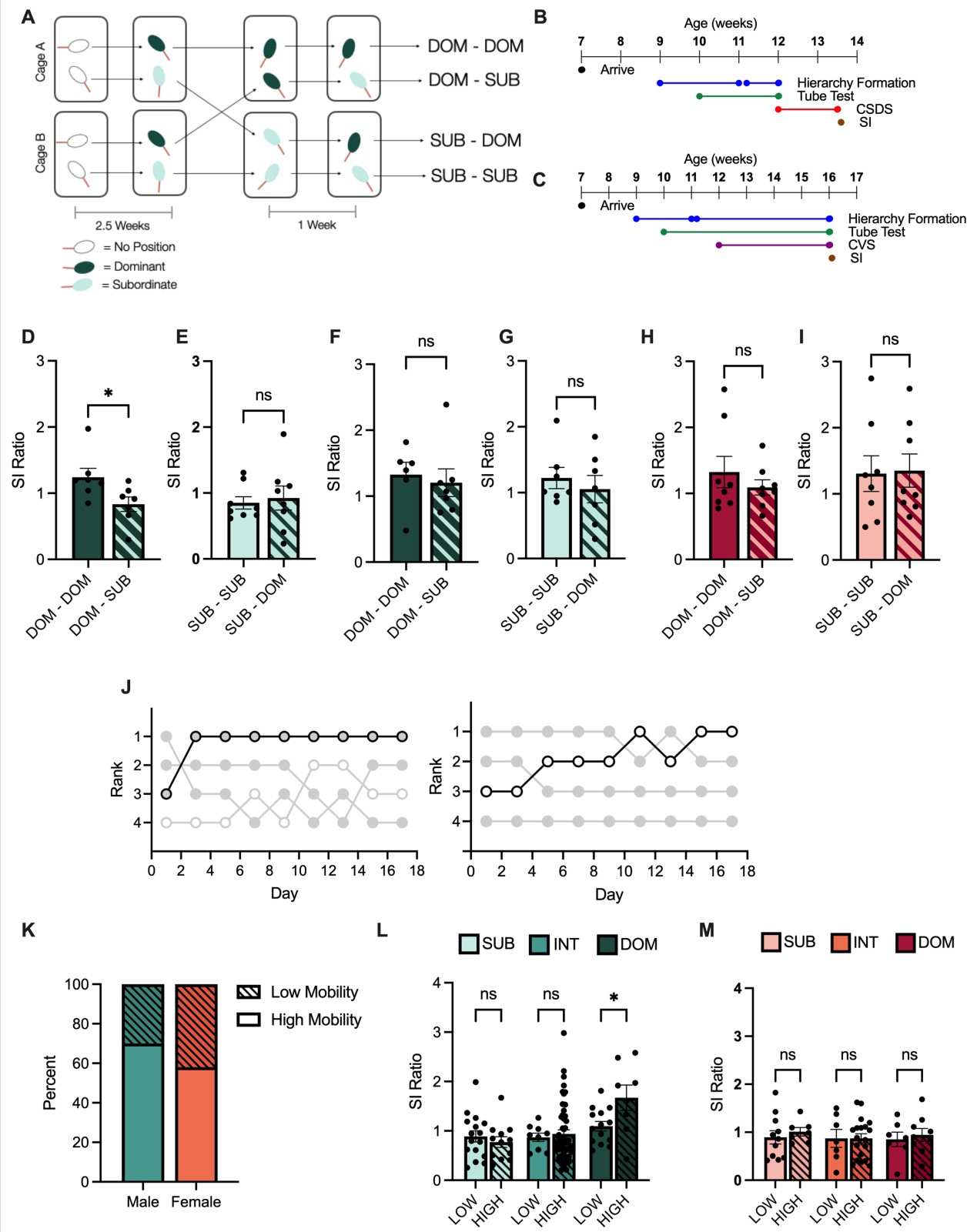

**Figure 4.** Manipulation of social hierarchy and individual history of winning. (**A**) Diagram of dyadic recombination. (**B**) Timeline of chronic social defeat stress (CSDS) dominance manipulation experiment. (**C**) Timeline of chronic variable stress (CVS) dominance manipulation experiment. (**D**) Social interaction (SI) ratio of initially dominant male mice following CSDS. t-test t(12) = 2.37, 95% CI [-0.78,–0.033], p = 0.0356 (n = 7/group). (**E**) SI ratio of initially subordinate male mice following CSDS. t-test t(14) = 0.36, 95% CI [–0.36, 0.51], p = 0.7239 (n = 7/group). (**F**) SI ratio of initially dominant male

*Figure 4 continued on next page*

*Figure 4 continued*

mice following CVS. t-test t(11) = 0.43, 95% CI [–0.75, 0.50], p = 0.6735 (n = 6/group). (**G**) SI ratio of initially subordinate male mice following CVS. t-test t(12) = 0.65, 95% CI [–0.74,.40], p = 0.5331 (n = 7/group). (**H**) SI ratio of initially dominant female mice following CVS. t-test t(14) = 0.88, 95% CI [–0.80, 0.33], p = 0.3937 (n = 8/group). (**I**) SI ratio of initially subordinate female mice following CVS. t-test t(14) = 0.12, 95% CI [–0.74, 0.83], p = 0.9042 (n = 8/group). (**J**) Representative traces of rank of mice with low mobility (left) versus high mobility (right). Low mobility ≤1 change in rank across hierarchy testing, high mobility ≥2 greater changes in rank across hierarchy testing. (**K**) Proportion of male (teal) and female (orange) mice demonstrating low mobility (striped) or high mobility (solid) hierarchies. $X^2$ (1, N = 159) = 1.7, p = 0.1408. (**L**) Social interaction (SI) ratio between low mobility (LOW) and high mobility (HIGH) male mice (mean ± SEM). Two-way ANOVA main Rank F(2, 104) = 7.65, p = 0.0008. Sidak's multiple comparisons of LOW versus HIGH within groups, SUB 95% CI [–0.36, 0.59] p = 0.9147 INT 95% CI [–0.52, 0.38] p = 0.9755 DOM 95% CI [-1.13,–0.02] p = 0.0387 (n = 9–16/group). (**M**) Social interaction (SI) ratio between LOW and HIGH female mice (mean ± SEM). Two-way ANOVA Rank F(2, 54) = 0.183, p = 0.6687 (n = 7–11/group). For all graphs *p ≤ 0.05, **p ≤ 0.01, ***p ≤ 0.001.

intrinsic characteristics, respectively, to determine rank. While female rank is highly transitive across cages, with outcomes against other mice sufficient to determine within cage rank, male rank is not. Furthermore, rigging of the tube test can form stable, alternate hierarchies in males, while not in females, suggesting that prior experience rather than intrinsic characteristics are utilized in males. Manipulation of gonadal sex was sufficient to reverse apparent strategies employed in hierarchy formation (*van den Berg et al., 2015*). While relying on different mechanisms to form their social hierarchies, we found that males and females had similar social hierarchies following a 3+ week period of formation with no sex-specific differences in linearity, stability, and distribution of DS. These hierarchies are generally linear on a day-to-day basis but dynamic over time, mirroring prior data that employed tube test to assign rank (*Varholick et al., 2018*). Furthermore, by using DS, we are able to capture aspects of the dynamic hierarchies and weight wins to the strength of the opponent. The positive correlation between rank assigned from DS and time in warm spot underscores the ability for DS calculated across a week of hierarchy testing to separate the ranks of both male and female mice in behaviorally distinct and ethologically relevant ways.

We employed the social interaction test to define stress susceptible and resilient animals. Here, susceptibility is defined by SI ratio below 1, which means that mice avoid interaction with a social target, while resiliency is defined by SI ratio over 1, which means mice spend more time with the social target. Resilient mice display social interaction levels similar to unstressed control mice. Concern in recent years over the validity of the SI test to evaluate stress response has been raised, as the target animals employed typically resemble aggressor animals from defeat. Thus, it is unclear whether social avoidance reflects a lack of social interest or whether it reflects generalization of fear responses to a perceived aggressor. However, by employing the SI test after both social and non-social stressors, we are able to more broadly assess the role of rank in stress response.

Broadly, we observed that rank is predictive of stress response to social stress enacted by the same sex, with dominance being associated with resilience and subordination associated with susceptibility. Dominant males were resilient to CSDS while intermediate and subordinate mice showed large deficits compared to their respective unstressed controls. In females, CSDS with a male aggressor caused social deficits across ranks, while inter-female CSDS negatively affected subordinates alone. This striking difference in stress outcomes across female stress models highlights the importance of testing across stress models. To probe the role of socially formed rank as a predictor of stress response, a social stress mirroring prior experience may be the most relevant to reveal social rank-associated differences.

Coping behavior during the final day of stress correlates with SI ratio across stress types and sex, suggesting that behavioral coping during stress can predict individual stress outcomes. Dominant male mice displayed higher levels of active coping behavior by day 10 compared to subordinate males, reflecting differences in SI ratio, while no differences were seen between dominant and subordinate females following CSDS. However, coping score still correlated with SI ratio across ranks in females. This suggests that coping score most closely reflects the states of susceptibility and resilience in the SI test, as subordinate females showed increased vigilance following inter-female CSDS but did not display reduced coping behavior during their final day of defeat, and vigilance did not correlate with coping. This retention of high levels of active behaviors in response to social aggression may also be reflective of retention of a dominant position by these mice, suggesting they have not yet assumed a subordinate position to the aggressor. Taken together, these results align with prior work investigating the impact of rank on coping behavior, where a positive relationship between active coping

behavior and reduced negative stress effects has been demonstrated (*Briand et al., 2015*; *Heshmati et al., 2018*).

Differences in stress experience do not account for rank-specific difference in stress responses. Given that mice of varied ranks may have different prior experience with antagonistic behavior, observations of aggressor behavior during defeat were necessary to rule out the possibility that differences in defeat experience were driving the rank-specific stress effects. We observed no differences between ranks in agonistic behavior received by the experimental animals either in direct measures, aggressive bouts experienced, or indirect measures, wounds received during social defeat. The lack of differences across ranks is likely due to the pathological nature of the aggression displayed by the aggressors that have been selected for extremely high levels of aggression. These results indicate that the differences seen in stress responses are not due to differences in the stress experienced by ranks, but rather are likely reflecting internal differences in response to the social stress.

In males, individual history of winning impacts stress outcome, while history of winning and manipulations of social rank do not affect stress outcomes in females. In males, high mobility dominants have a strikingly higher SI ratio following social stress than low mobility dominants, suggesting that experiencing mobility may be stress protective. These results may seem unexpected given the primate literature, which has indicated that dominant animals from a stable social hierarchy are less stressed than dominant animals from an unstable social hierarchy (*Abbott et al., 2003*; *Sapolsky, 2005*). However, in our study, both high and low mobility dominant mice originated from a mix of stable and unstable cages, indicating that rank stability of dominant mice is not indicative of the overall hierarchy landscape. Furthermore, while some cages displayed periods of identical consecutive tube test results, in no cases were any social hierarchies completely stable over all 18 days of testing.

The difference between high and low mobility dominant males can be interpreted in several ways. One interpretation is that low mobility dominants were placed in a cage of relatively subordinate animals and did not need to be particularly dominant to retain their position, and therefore were indistinguishable from other ranks when measuring later social stress susceptibility. In contrast, high mobility dominants formed their rank in a highly competitive environment against other dominant animals and thus, the mice that rose to the top are especially dominant, which confers higher levels of stress resilience. Another possibility is that high mobility dominants, while successfully rising to the top of their respective hierarchies, benefited from the winner effect conferring them with biological advantages that set them up for future success (*Hsu and Wolf, 1999*). These advantages could bias dominant animals to continue to fight for a dominant position longer than less dominant animals, and thus protect them from the stress of loss of rank conferred by social defeat. These interpretations are not mutually exclusive and could be occurring in parallel.

Intriguingly, there are no differences in stress resilience between low and high mobility females across ranks. This lack of difference could be due to the reliance of females on intrinsic characteristics to determine social rank position. This is supported by the fact that the winner effect has only been weakly demonstrated in less aggressive strains of mice, unless supplemented with testosterone following a win (*Fuxjager et al., 2011*). Therefore, winning experience in females may not confer advantage when exposed to later physical aggression during social defeat. Additional work to disentangle the contributions of innate stress susceptibility versus learned resilience in both males and females remains to be completed. However, these results highlight how the history of social rank of a male mouse can be indicative of divergent stress outcomes, especially within the dominant rank. Therefore, our data highlights the need to consider hierarchy formation preceding stress exposure when interpreting the predictive abilities of rank on stress susceptibility.

The importance of accounting for the hierarchy landscape is further underscored by the lack of differences in stress susceptibility observed between recombined subordinate males or females following CSDS or CVS. SUB - DOM and SUB - SUB animals are indistinguishable in stress outcomes. This is seen in response to both CSDS, which occurred after 1 week of recombination, or CVS which occurred after 5 weeks of recombination, suggesting that giving more time to adjust to a newly dominant position would not increase a previously subordinates animal's stress resilience. This suggests that initial predisposition for subordination confers susceptibility to stress and it is not from stress experienced while occupying the subordinate rank. This would also align with prior results which found no significant differences between tube test-ranked male C57BL/6 mice in bodyweight, locomotion, anxiety or corticosterone either as preexisting measures or those taken after establishment of hierarchy (*Pallé*

*et al., 2019*). It follows then that a subordinate rank position in a non-competitive environment is not associated with increased stress levels. It is only under high levels of competition, such as in the visible burrow system or frequent cage changes to escalate agonistic behavior, where subordination is stressful and subordinate animals start to show behavioral and biological consequences of stress exposure (*Blanchard et al., 1995*; *Horii et al., 2017*; *Melhorn et al., 2017*; *Tamashiro et al., 2007*).

In contrast to subordinate animals where recombination does not alter stress susceptibility, a dominant male must maintain a dominant position to retain stress resilience; DOM - SUBs are significantly more susceptible than DOM - DOMs to social stress alone. This result mirrors prior work in hamsters revealing dominance maintenance is necessary to maintain stress resilience (*Morrison et al., 2014*). This effect could be due to the stress of recently losing a social hierarchy position, or because the DOM - SUB animal is inherently less dominant, and thus less resilient, than the other animal. Manipulation before CVS does not cause significant differences between DOM - DOMs and DOM - SUBs in both males and females, suggesting that resilience to social stress seen in dominants may rely on learned experiences during hierarchy formation. Separating the contributions between innate versus acquired stress resilience is a challenging issue that requires additional investigation. Clearly, a dominant animal is not always equivalent to a dominant animal from another cage and therefore accounting for the social landscape in which that dominant animal rose to and subsequently maintained their position is vital.

Our findings stand in contrast to prior work which revealed an opposite pattern of resilience by rank. Previously it has been demonstrated that the most dominant 50 % of male mice are more stress susceptible to social defeat (*Larrieu et al., 2017*), while others have shown no relationship between rank and social stress susceptibility (*Šabanović et al., 2020*). Closer examination of the social landscape reveals potential explanations for these differences. The study by *Larrieu et al., 2017* were performed on a strikingly different social landscape where a majority of cages (90%) had completely stable social hierarchies across all 6 days of testing. Because these hierarchies were completely stable, animals were divided by absolute rank within cage, unlike our divisions which utilized DS calculated over the preceding week to capture aspects of the animal's dynamic history of winning. Under completely stable social conditions, additional factors may influence both hierarchy formation and social stress responses. This has been demonstrated in humans where under stable or unstable hierarchy situations, different neural regions are engaged when viewing a superior (*Zink et al., 2008*). The level of despotism of dominant animals and overall levels of agonistic behavior in an established hierarchy can also interact to differentially affect behaviors and hormone levels of dominant and subordinate rodents (*Varholick et al., 2019*; *Williamson et al., 2017*). An additional factor that likely shapes the relationship between rank and stress susceptibility is the length of group housing prior to testing. *Šabanović et al., 2020* group-housed mice for 2 weeks before stress exposure, and in this case, they did not find any differences in stress susceptibility between ranks. This result in conjunction with ours importantly argues against innate resilience in dominants for males, but instead suggests there are advantages gained from both acquiring and maintaining a dominant position. However, direct comparison between this research is challenging, as they rely on both different means of testing and categorizing dominant and subordinate animals.

Environmental stress may also serve as an external factor shaping hierarchy formation or may be acting in combination with stress experience related to hierarchy mobility within the cage. In the case of our CVS data, rank derived from the week prior to stress initiation was insufficient to predict social interaction deficits. Instead, we only reveal susceptibility in both sexes when rank is calculated across the final week of stress. Therefore, it may be the case that rank position assumed under stress is due to the animal's response to the stress experienced. Dominant animals, which have a smaller stress response to the variable stress, remain in a dominant position, whereas animals with larger stress responses drop to the bottom of the social hierarchy. This effect may take time as a single acute stressor did not alter hierarchy when measuring agonistic behavior in a social box (*Karamihalev et al., 2020*). In a reverse manner, it may be that subordinate rank associated social avoidance following CVS is reflective of the increased stress associated with acquiring a lower social position. This is supported by the data which shows that previously susceptible animals can still achieve a dominant position within a cage of novel animals; however external environmental stress may still serve to shape the continued dynamics of an established social hierarchy (*Šabanović et al., 2020*). Thus, close examination of social hierarchy position across time is vital in predicting animals' stress susceptibility.

In this study, we have demonstrated a significant role for social rank as a predictor of later stress susceptibility to both social and non-social stress in male and female mice. In both stress domains, dominant rank was associated with resilience and subordinate rank with susceptibility. Furthermore, in males we have highlighted the importance of accounting for not only an individual's rank, but that individual's history of winning as well as the social landscape under which rank was formed. Rank-derived stress susceptibility or resilience is likely a dynamic interplay between pre-existing disposition, learned behavioral responses, biological responses to external factors, and the overall social landscape. Further work to tease out the exact biological underpinnings of each of these elements will be crucial to fully understand the link between social rank and stress vulnerability.

# Materials and methods

**Key resources table**

| Reagent type (species) or resource | Designation | Source or reference | Identifiers | Additional information |
|---|---|---|---|---|
| Strain, strain background (*Mus musculus*) | C57 Mouse (Male and Female) | The Jackson Laboratory | C57BL/6 J | RRID:IMSR_JAX:000664 |
| Strain, strain background (*Mus musculus*) | CD-1 (Male) | Charles River | Crl:CD1(ICR) | RRID:IMSR_CRL:022 |
| Strain, strain background (*Mus musculus*) | Esr1-cre (Male) | The Jackson Laboratory | C57BL/6-Esr1$^{tm2.1(cre)And}$/J | RRID:IMSR_JAX:017913 |
| Strain, strain background (*Mus musculus*) | Swiss Webster (Female) | Charles River | Crl:CFW(SW) | RRID:IMSR_CRL:024 |
| Software, algorithm | Ethovision | Noldus Information Technology | | Version 11.0 |

## Animal housing and care

All animal procedures were approved by the Icahn School of Medicine at Mount Sinai Institutional Animal Care and Use Committee (Protocol #: LA10-00266 to S.J.R.) All mice were provided ad libitum access to food and water and maintained on a 12 hour light/dark cycle. All behavioral testing was performed during the light cycle.

### Experimental
Seven-week-old male and female C57BL/6 J RRID:IMSR_JAX:000664 mice were randomly combined into tetrads (4/cage) of same sex mates and allowed to cohabitate for 3 weeks prior to initiation of hierarchy testing.

### Male aggressors for male CSDS
Retired male CD-1 breeders RRID:IMSR_CRL:022 between 3 and 6 months old were acquired and maintained in single housing following 3 days of screening for sufficient levels of inter-male aggressive behavior, as previously described (*Golden et al., 2011*).

### Male aggressors for female CSDS
Male heterozygous *Esr1*-Cre mice derived from RRID:IMSR_JAX:017913 were injected bilaterally with Cre-dependent AAV-DIO-Gq-DREADD into the ventrolateral subdivision of the ventromedial hypothalamus (VMHvl) and subsequently screened for sufficient CNO-triggered aggressive behavior towards female mice as previously described, and maintained in single housing (*Takahashi et al., 2017*).

### Female aggressors for female CSDS
12 week or older female Swiss Webster RRID:IMSR_CRL:024 mice were housed with castrated male mice and screened for inter-female aggressive behavior as previously described (*Newman et al., 2019*).

## Hierarchy testing

### Tube test

This protocol was adapted from *Fan et al., 2019*; *Lindzey et al., 1961*. At 10 weeks of age, 3 days prior to hierarchy evaluation, mice were habituated to handling for 5 min per mouse. On the second day of habituation, cages of mice were allowed to explore the testing arena, a rat cage (26.7 cm width ×48.3 cm depth ×15.2 cm height, Allentown Inc) with a plexiglass tube (3 cm internal diameter x 30 cm length) taped to the bottom. On the final day of training, mice were trained to pass through the tube for a minimum of 5 times in each direction, followed by 20 s of arena investigation. Mice were not permitted to escape backwards out of the tube. The tube was cleaned and dried in between trials. Starting at 10 weeks every other day for 18 days mice were pitted against cagemates in a round robin design randomly ordered. For each trial, two mice were guided simultaneously to opposing ends of the tube and permitted to meet in the middle. The trial ended when one mouse backed up out of the tube ending with all four paws on the outside cage floor; this mouse was considered the trial 'loser' and the mouse remaining in the tube was the 'winner'. The winning mouse was left to leave the tube forwards and not permitted to back out of the tube. If both mice attempted to leave the tube backwards the trial was ended, both mice placed back into the home cage before initiating the trial again. For stressors where mice remained group housed (CVS and female CSDS), hierarchy evaluation was continued every other day for the duration of the stressor until mice were single housed prior to behavioral testing.

### Warm spot test

The warm spot test was adapted from the protocol as previously described (*Zhou et al., 2017*). The test was completed at 12 weeks of age. In brief, an empty housing cage (26.7 cm width ×48.3 cm depth) was cooled on ice until the floor reached 3°C. Under one corner of the cage, a waterproof heating pad was placed to locally heat the floor to 30°C. For male mice, a small plastic hut sufficient to hold a single male mouse was placed over the warm corner. For female mice, no hut was employed. Thirty min prior to trial initiation, mice were transferred from their home cage to an empty cage maintained on ice to cool down. For each trial, mice were placed in the test cage, and behavior was tracked for 20 min for males and 10 min for females by video-tape. For male mice occupation time of the hut was measured, for female mice, primary occupation of the warm corner was measured.

### Calculating dominance

Dominance was evaluated via David's Score (DS) as previously described (*David, 1987*; *Gammell et al., 2003*). DS is a weighted measure based on results from paired competitions. DS was calculated from the four tube test results preceding stress start, a period spanning 8 days. This approach was chosen because all mice were included in future behavioral testing regardless of original cage hierarchy stability (defined as four consecutive days of identical tube test results.) Dominants were the top 25 % of DS (DS >3), subordinates were the bottom 25 % (DS $\leq$ –3), and intermediates were the central 50 % (-3< DS $\leq$ 3). High and low mobility were defined as upwards or downwards change in hierarchy position based on position from every days' linear tube test results. Low mobility animals were those with one or fewer changes in rank, whereas high mobility animals were those with two or greater changes in rank. Rank change was calculated over all days preceding stress initiation. Steepness was calculated over seven tube test results (spanning 2 weeks) before or after stress initiation. Steepness is the slope of the linear regression line between ordered normalized DSs. Cages of mice were randomly assigned to stress or control groups for future manipulations.

### Hierarchy manipulation in dyads

Male or Female C57BL/6 J mice were recombined into weight-matched dyads ( ± 0.5 grams) with an unfamiliar partner at 9 weeks of age. Following 2 weeks of hierarchy formation and 1 week of hierarchy testing (via tube test), dominant or subordinate mice were recombined with a weight-matched unfamiliar mouse that was of matching hierarchy position. These new dyads were then tested for hierarchy for 1 week before being placed into stress. Animals that were initially dominant and then remained dominant following recombination were called 'DOM-DOM', those that started dominant and become subordinate were called 'DOM-SUB'. Mice that were initially subordinate and remained

subordinate were termed 'SUB-SUB', and those that were initially subordinate but become dominant were termed 'SUB-DOM'.

## Male CSDS

Male chronic social defeat stress (CSDS) was performed as previously described at 13 weeks of age (*Golden et al., 2011*). At least 48 hr prior to the start of social defeat, male aggressors were rehoused into the social defeat cage (26.7 cm width ×48.3 cm depth ×15.2 cm height, Allentown Inc) divided in half by a clear perforated divider (0.6 cm × 45.7 cm × 15.2 cm, Nationwide Plastics). Experimental animals were subjected to agonistic encounters from a novel aggressor for 10 minutes every day for 10 consecutive days. Following this encounter, experimental mice were placed across the plastic divider from the aggressor for 24 hr of sensory exposure. Experimental animals were single housed following the final bout of defeat on the 10th day prior to social interaction testing, and the social interaction test was completed 24 hr later.

## Female CSDS

Female chronic social defeat stress was performed as previously described at 13 weeks of age (*Takahashi et al., 2017*). DREADD aggressors received an intraperitoneal injection of 1.0 mg/kg clozapine-N-oxide (CNO) 30 minutes prior to the start of the defeat. Individual experimental animals were introduced into a novel aggressor's cage for 5 min from the start of the first agonistic encounter for 10 consecutive days before being returned to their original cage. Control and stressed animals remained group housed during the defeat and were single housed following the final bout of defeat on the 10th day prior to the social interaction test, which was completed 24 hr later.

## Interfemale CSDS

Interfemale CSDS was completed as previously described at 13 weeks of age (*Newman et al., 2019*). 48 hours before social defeat, aggressive females housed with castrated males were rehoused into the social defeat cages (26.7 cm width ×48.3 cm depth ×15.2 cm height, Allentown Inc) divided in half by a clear perforated divider (0.6 cm × 45.7 cm × 15.2 cm, Nationwide Plastics). Male mates were temporarily removed and experimental female mice were introduced to the novel aggressive female's cage for 5 min each day for 10 consecutive days. Following physical defeat, experimental females were moved across the clear plastic divider for 24 hr of sensory exposure. Every other day fresh bedding was introduced into the cage, and on the 5th day of defeat, all defeat cages were changed to maintain female aggressor levels. Control animals were single housed across from cage mates and were rotated every other day to a new cage mate. Animals were single housed following the final bout of defeat on the 10th day prior to social interaction testing, which was completed 24 hr later. Vigilance was scored as the time an animal spent directed towards the aggressor outside of the interaction zone.

## Male and female CVS

CVS was administered as previously described at 13 weeks of age (*Labonté et al., 2017*). Stressed animals were exposed to 21 days of daily 1 hr stressors consisting of either 100 random mild foot shocks at 0.45 mA (administered to a single cage of mice at a time), tail suspension or restraint stress in a 50 mL falcon tube. Stressed and unstressed controls were group housed for the duration of the stress and were single housed before the initiation of the behavioral testing.

## Social interaction test

Social interaction testing was completed as previously described at ~14.5 weeks of age (*Golden et al., 2011*). The social interaction test comprises two trials: a 2.5 min target absent trial where experimental animals explore an open field arena with an empty wire and plexiglass chamber, followed by a 2.5 min target present trial where novel aggressors are introduced to the chamber. Animals' movement including time spent in the interaction zone was tracked via Ethovision (Version 11.0 Noldus Information Technology Inc, Leesburg, VA) RRID:SCR_000441. Social interaction ratio (SI ratio) was calculated as the time spent in the interaction zone when target was present divided by when target was absent. The arena and chamber were cleaned and dried between mice. Novel target mice were as listed: male CSDS: male CD1 aggressors, female CSDS: DREADD Aggressors without CNO injection, inter-female CSDS: female Swiss Webster aggressors, CVS: male CD1 aggressor.

## CSDS behavioral coping scoring

Behaviors exhibited by the experimental mice during defeat were scored during post hoc video analysis by a blinded experimenter on days 1 and 10 of defeat. Active behaviors were defined as running to the opposing end of the cage after attack initiation by the aggressor, and biting, boxing or lunging at the aggressor. Passive behavior was defined as immobility in response to attack initiation. An attack bout was defined as a discrete aggressive encounter in which the aggressor was in constant contact or immediately chased after the experimental animal. The active coping score was calculated from the total number of active behaviors displayed, minus the total number of passive behaviors normalized to the total number of aggressive bouts the experimental animal was exposed to (Escape+ Fighting back – Passive / # Attacks).

## Wound score

Wounding following social defeat was calculated 48 hr following the final day of defeat. Wound score is a summed number from a three point system across 4 areas of the body (lower back, upper back, tail, and abdominal) with 0 points in all categories equivalent to no wounds. Lower back points are assigned by wound area ($0.01–1\ cm^2$ = 1 point, $1–2\ cm^2$ = two points, and greater than $2\ cm^2$ = 3 points). All other areas are counted as total number of discrete wounds (1–5 = 1 point, 5–10 = 2 points, 10+ = 3 points).

## Statistics

All statistical tests were performed in Prism (Version 9.1.1; Graphpad) RRID:SCR_002798. Outliers, defined as two standard deviations from the mean within each rank and stress condition, were removed. Data were tested for normality by Lavene's test for homogeneity of variance. All results are reported as means ± SEM. Power Analyses for CVS + Manipulation experiments were completed with the 'pwr' package in R (Version 3.5.1) RRID:SCR_001905 utilizing error variance and mean differences in stressed condition in the social defeat experiments.

---

# Additional information

### Funding

| Funder | Grant reference number | Author |
|---|---|---|
| National Institutes of Health | R01MH114882 | Scott J Russo |
| National Institutes of Health | R01MH127820 | Scott J Russo |
| National Institutes of Health | R01MH104559 | Scott J Russo |
| CAPES-Brazil | Visiting Researcher Fellowship | Manuella P Kaster |
| Canadian Institutes of Health Research | Postdoctoral Fellowship 201811MFE-414896-231226 | Kenny L Chan |
| Leon Levy Foundation | Postdoctoral Fellowship | Lyonna F Parise |

The funders had no role in study design, data collection and interpretation, or the decision to submit the work for publication.

### Author contributions

Katherine B LeClair, Conceptualization, Data curation, Formal analysis, Investigation, Methodology, Project administration, Validation, Visualization, Writing - original draft, Writing - review and editing; Kenny L Chan, Conceptualization, Formal analysis, Investigation, Methodology, Writing - review and editing; Manuella P Kaster, Conceptualization, Investigation, Methodology; Lyonna F Parise, Conceptualization, Investigation, Methodology, Writing - review and editing; Charles Joseph Burnett,

Investigation, Methodology, Writing - review and editing; Scott J Russo, Conceptualization, Funding acquisition, Methodology, Resources, Supervision, Writing - original draft, Writing - review and editing

**Author ORCIDs**
Katherine B LeClair http://orcid.org/0000-0003-2075-7740
Scott J Russo http://orcid.org/0000-0002-6470-1805

**Ethics**
All animal procedures were approved by the Icahn School of Medicine at Mount Sinai Institutional Animal Care and Use Committee (Protocol #: LA10-00266 to SJR).

**Decision letter and Author response**
Decision letter https://doi.org/10.7554/eLife.71401.sa1
Author response https://doi.org/10.7554/eLife.71401.sa2

## Additional files

**Supplementary files**
• Transparent reporting form

**Data availability**
All data generated or analyzed during this study are included in the manuscript.

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
