## [Decision Letter]

**Acceptance summary:**

This manuscript represents an elegant and thorough examination of how social hierarchy influences stress responsivity in both male and female mice, including a thoughtful discussion of how the work fits into current models and potential mechanisms underlying the observations.

**Decision letter after peer review:**

Thank you for submitting your article "History of winning and hierarchy landscape influence stress susceptibility in mice" for consideration by *eLife*. Your article has been reviewed by 2 peer reviewers, one of whom is a member of our Board of Reviewing Editors, and the evaluation has been overseen by Kate Wassum as the Senior Editor. The following individual involved in review of your submission has agreed to reveal their identity: Carmen Sandi (Reviewer #2).

Essential revisions:

1) In some cases the stats are not appropriate, for example, why did the authors run a correlation for Figures 1E-F? With 3 groups (not individual animal data points), a 1-way ANOVA is the correct analysis. This should be fixed.

2) Although the social interaction test has been widely used to define stress vulnerability, there have also been several detractors as to its validity in the context of the CSDS model. It would be important that the authors elaborate on this point, how to overcome this critique in the Discussion section. This point is particularly relevant in the context of this study in which social hierarchy is challenged and animals have been exposed to several winning/losing experience/s.

3) In relation to the previous point, it would be important to discuss the concepts of "susceptibility" and "resilience" in the context of the current experimental design and study.

4) Finally, it would be relevant to discuss the current findings in the context of earlier proposals in the literature (PMID: 29869396) suggesting that, beyond to rank, it may be the actual consequences of winning or losing what impacts individuals' health.

5) If you have not already done so, please ensure your manuscript complies with the *eLife* policies for statistical reporting: https://reviewer.elifesciences.org/author-guide/full "Report exact p-values wherever possible alongside the summary statistics and 95% confidence intervals. These should be reported for all key questions and not only when the p-value is less than 0.05." This includes full reporting of F values and degrees of freedom for ANOVAs.

---

## [Author Response]

Essential revisions:1) In some cases the stats are not appropriate, for example, why did the authors run a correlation for Figures 1E-F? With 3 groups (not individual animal data points), a 1-way ANOVA is the correct analysis. This should be fixed.

We appreciate the reviewer’s attention to detail and have updated the graphs in figures 1E – F. In one of the original publications employing the warm corner test, a similar analysis was employed looking at the relationship between rank and time spent in the warm corner (Zhou et al., 2017). While rank is categorical it is also ordinal, therefore application of a spearman regression across all individual data points is appropriate. We agree that the original visualization showing average + SEM overlaid with the regression line was difficult to interpret and have updated the graph to display individual data points. In addition, we have included a bar graph and performed a one-way ANOVA to enhance the discussion regarding the exact relationship between time spent in the warm spot and each rank.

2) Although the social interaction test has been widely used to define stress vulnerability, there have also been several detractors as to its validity in the context of the CSDS model. It would be important that the authors elaborate on this point, how to overcome this critique in the Discussion section. This point is particularly relevant in the context of this study in which social hierarchy is challenged and animals have been exposed to several winning/losing experience/s.

The reviewer raises an important point, which we address with additional discussion about the use of social interaction as a behavioral test to determine susceptibility and resilience following CSDS. We have highlighted the importance of testing across stress models on lines 306 – 308. As well, we have underscored the necessity for evaluating aggressor behavior during the defeat on lines 338 – 341 to eliminate alternate explanations for differences in SI behavior.

3) In relation to the previous point, it would be important to discuss the concepts of "susceptibility" and "resilience" in the context of the current experimental design and study.

In response to the reviewer’s point, we have incorporated more direct discussion of operational definitions of susceptibility versus resilience and how they relate to our current experimental design on lines 297 – 303.

4) Finally, it would be relevant to discuss the current findings in the context of earlier proposals in the literature (PMID: 29869396) suggesting that, beyond to rank, it may be the actual consequences of winning or losing what impacts individuals' health.

We agree with the reviewer that placing the results of this paper in the context of prior proposals in the literature, specifically the role rank loss contributes to stress of social defeat, is highly relevant. To address this, we have incorporated additional discussion of the existing literature, which can be found on lines 269 – 271, 338 – 341, 371 – 373, 460-462.

5) If you have not already done so, please ensure your manuscript complies with the eLife policies for statistical reporting: https://reviewer.elifesciences.org/author-guide/full "Report exact p-values wherever possible alongside the summary statistics and 95% confidence intervals. These should be reported for all key questions and not only when the p-value is less than 0.05." This includes full reporting of F values and degrees of freedom for ANOVAs.

F Values and degrees of Freedom for ANOVA, as well as 95% confidence intervals have now been included.